# Frequent *Klebsiella pneumoniae* Urinary Tract Infections in a Patient Treated with Ruxolitinib

**DOI:** 10.3390/antibiotics8030150

**Published:** 2019-09-16

**Authors:** Ramy M. Hanna, Maham Khalid, Lama Abd El-Nour, Umut Selamet

**Affiliations:** 1Division of Nephrology, Department of Medicine UCLA, David Geffen School of Medicine, University of California, Los Angeles, CA 90095, USA; labdelnour@mednet.ucla.edu (L.A.E.-N.); uselamet@hotmail.com (U.S.); 2Division of Nephrology, Department of Medicine, UCI School of Medicine, Irvine, CA 92868, USA; maham.khalid@yahoo.com; 3Division of Nephrology, Department of Medicine, Brigham Women’s and Children’s Hospital, Boston, MA 02101, USA

**Keywords:** Ruxolitinib, *Klebsiella pneumonia*, Immunemodulation, JAK kinase (Janus Kinase), STAT (Signal Transducer and Activator of Transcription proteins) kinase

## Abstract

Ruxolitinib is a targeted agent that inhibits Janus 2 Kinase and is approved for use in Polycythemia Vera and Primary Myelofibrosis. Its mechanism of action involves inhibition of cellular proliferation via the Janus kinase/signal transducer and activator of transcription proteins pathway. Ruxolitinib has different immune modulating effects that result in functional immunosuppression, leading to an increased susceptibility to certain infections. *Klebsiella pneumoniae* infections, in particular, were common among the reported pathogens contracted by ruxolitinib users. We report a 75-year-old male patient who had recurrent *K. pneumoniae* urinary tract infections while on ruxolitinib for Polycythemia Vera. This case is reported to add to the literature describing an increased susceptibility of patients to this often-resistant bacteria and to raise awareness about the immune modulating effects of JAK inhibitors.

## 1. Introduction

Ruxolitinib is an oral Janus Kinase (JAK) 1 and 2 inhibitor that is part of the family of tyrosine kinase inhibitors (TKI) [1]. This agent is approved by the US Food and Drug administration for the treatment of Polycythemia Vera (PCV) and Primary Myelofibrosis (PMF) [2]. While a rare condition, (2.8/100,000 in males, 1.3/100,000 in females), PCV had few modern therapy options with phlebotomy still being used as an option [3]. The efficacy of Ruxolitinib has been established in the Ruxolitinib versus Standard Therapy for the Treatment of Polycythemia Vera (RESPONSE) for PCV [2,4], and Controlled MyeloFibrosis Study with ORal JAK inhibitor Treatment I and II (COMFORT I and II) for myelofibrosis [2]. It is also being used in hemophagocytic lymphohistiocytosis [5].

The JAK/STAT (Signal Transducer and Activator of Transcription proteins) pathway is intimately involved with cellular division in hematopetic progenitor cell division and proliferation of adaptive immunity T cells [6]. Given its role in many dividing cells of the immune system, it can be expected to have an immune modulatory effect [7].

Evidence of this effect emerged when many patients using ruxolitinib were reported to have an increased susceptibility to bacterial infections. This was first reported in myelofibrosis patients [6], but later was also found to be true in PCV patients [8]. In addition to increased susceptibility to bacterial infections, increased risk of fungal [9] and viral infections were subsequently reported [9]. The cells affected by ruxolitinib include T cells, natural killer cells, and dendritic cells [10]. One particular bacterial pathogen that has been reported in association with Ruxolitinib use has been *K. pneumoniae*, with the best-known reported patient developing a hepatic abscess [11,12]. We report another patient on ruxolitinib who developed frequent *K. Pneumoinae* urinary tract infections, given these reports, this case is documented to show the clinical link between ruxolitinib use and infection with an often resistant and virulently pathogenic organism.

### Case Report

Our patient is a 75-year-old Caucasian male who was diagnosed with PCV that transitioned to myelofibrosis. He also had a history of non-ischemic cardiomyopathy with an ejection fraction of 30%, who was treated with ruxolitinib 20 mg by mouth twice a day. He presented to nephrology care for monitoring for myelofibrosis related nephropathy, which was suppressed due to angiotensin converting enzyme (ACE) inhibitor use. The patient’s serum creatinine improved when he was instructed to avoid non-steroidal anti-inflammatory agents. His serum potassium rose despite an improving serum creatinine and he had to be taken off his ACE inhibitors. His serum creatinine ranged between 1.3–1.7 mg/dL on ACE inhibitor and afterwards dropped to 1.2–1.3 mg/dL. The estimated glomerular filtration rate of 40 mL/min improved to 54 mL/min after ACE inhibitor discontinuation.

He had some proteinuria at 0.6 grams protein/gram creatinine but further investigation of his urine with a urinary culture being sent revealed a urinary tract infection and the patient noted frequency symptoms. His first infection was noted 12/2018 he was treated with a course of ciprofloxacin. His urinary culture returned with only one organism: >100,000 colony forming units (cfus) of *K. pneumoniae.* This isolate was resistant to ampicillin, susceptible with a minimum inhibitory concentration (MIC) of ≤1 µg/mL ceftriaxone, ≤0.25 µg/mL for ciprofloxacin, ≤1 µg/mL for gentamicin, ≤16 µg/mL for nitrofurantoin, susceptible to oral cephalosporins, ≤4 µg/mL for piperacillin/tazobactam, and ≤20 µg/mL for trimethoprim/sulfamethoxazole. Despite a week long course of ciprofloxacin 250 mg orally twice a day for a week, the patient reported ongoing symptoms. In 2/2019, a repeat urinary culture was drawn. This culture showed one organism, again >100,000 cfu of *K. pneumoniae*, with the same sensitivity pattern as the 12/2018 isolate except for ≤32 µg/mL MIC for nitrofurantoin. This time the patient was treated with cephalexin 500 mg by mouth for 1 week, and experienced some relief from symptoms. One month later, 3/2019, the patient reported urinary symptoms again and a similar isolate of *K. pneumoniae* was found with the same sensitivity pattern except now it was resistant to ampicillin and nitrofurantoin. This final time the patient was treated with cefuroxime 250 mg orally twice for 10 days. He finally reported resolution of urinary frequency, but is being monitored for recurrent symptoms and infection. The patient had no indwelling catheters, instrumentalization, kidney stones, or any other reason for a complicated urinary tract infection. It was felt clinically the patient may be having trouble clearing his *K. pneumoniae* infection, and the literature discussed earlier regarding ruxolitinib was discussed as a possible explanation for these recurrent urinary tract infections (UTIs).

## 2. Discussion

The nearly ubiquitous role of tyrosine kinase inhibitors in cell division across red cell and immune cell progenitors provide a clear context for the classification of ruxolitinib as an immune modulating or even immunosuppressing agent [6,7,8,9,10,11,12]. Figure 1 shows the JAK/STAT kinase pathway as a schematic drawing. It is noted that ruxolitinib was postulated as a main etiology for why the reported patient could not clear his underlying *K. Pneumoinae.* urinary tract infections in this case. It is important in PCV, PMF, and other patients receiving this agent that the potential for viral, fungal, and bacterial infections [6,7,8,9,10] be appreciated and the patients should be monitored accordingly.

While most infections associated with ruxolitinib are mild and treated successfully with antibiotics [1,2], the potential for rare [9], fungal [9], viral [8], and bacterial [6,7,8,10,11,12] infections that may present with abscesses, non-clearing infections, or even life threatening ways must be recognized and anticipated [10,11,12,13]. Please see Table 1 for the three reported cases of *K. pneumoniae* infections in patients receiving ruxolitinib.

It is important for oncologists, onconephrologists, and infectious disease specialists to recognize the immune suppressing effects of ruxolitinib, and the predilection for various atypical organisms [6,7,8,10,11,12,13], including the risk for the often resistant *K. pneumoniae* species. Given immunomodulatory/immune suppression risk of JAK 1,2 inhibitors, the risk of atypical and clinically silent infections must also be kept in mind when evaluating the patient on ruxolitinib. The beneficial effects of ruxolitinib on patients with myelofibrosis and PCV also warrant continued use of this agent, but with a vigilant eye for increased infection risks [14].

## 3. Conclusions 

This report is to increase awareness of ruxolitinib as an immunomodulatory/immuno-suppressive agent. *Klebsiella pneumoniae* infection can be seen with JAK kinase agents like ruxolitinib. Reported cases include recurrent urinary tract infections and one case of a hepatic abscess with *K. pneumoniae* in users of ruxolitinib.

## Figures and Tables

**Figure 1 antibiotics-08-00150-f001:**
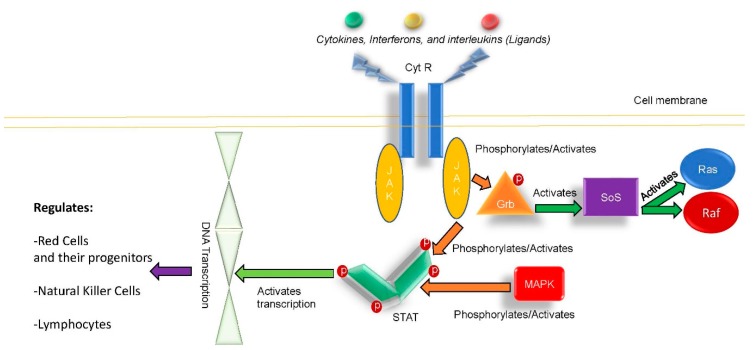
Schematic of Janus kinase 2 pathway (JAK/STAT pathway). CytR, cytokine receptor; DNA, deoxyribonucleic acid; Grb, Growth factor receptor-bound protein 2; JAK-Janus Kinase 1, 2; MAPK, a mitogen-activated protein kinase; P, phosphorous; Ras, rat sarcoma protein; Raf, serine/threonine kinase/cellular homolog of viral RAF gene; SoS, son of sevenless; STAT, signal transducer and activator of transcription.

**Table 1 antibiotics-08-00150-t001:** Cases of *K. Pneumoniae* infections associated with ruxolitinib use.

Reference	Age	Gender	Race	Indication	Infection Location
12	78	M	NR	PMF	Liver abscess
13	NR	NR	NR	NR	UTI
CC	75	M	Caucasian	PCV	UTI

CC, current case; M, male; NR, not reported; PCV, polycythemia vera, PMF, primary myelofibrosis; UTI; urinary tract infection.

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
