# Peer review of "Frequent Klebsiella pneumoniae Urinary Tract Infections in a Patient Treated with Ruxolitinib"

_antibiotics, 2019, doi:10.3390/antibiotics8030150_

Round 1

Reviewer 1 Report

In this manuscript is reported a case involving the possible association between the use of Ruxolitinib and increased frequency of urinary infection caused by Klebsiella pneumoniae.

As it is known that this drug increases the susceptibility to infections and therefore the reported possible association is possible, there is no marked novelty of this work. However, I agree with the authors in that “This case is reported to add to the literature describing an increased susceptibility of patients to this often-resistant bacteria and to raise awareness about the immune modulating effects of JAK inhibitors”. Thus, I think that this document can be considered for publication.

However, several mistakes can be detected:

1) Please correct the name of the bacterium to “Klebsiella pneumoniae” (including in italics), as mistakes can be detected over all the text.

2) Keywords:

-need formatting

-mistakes: “Klebesiella” and “Immune modulation”

3) Abstract:

-lines 22/23: please correct sentence

-please define the abbreviation STAT

-PCV was defined twice

4) Main text:

Lines 42/43: please correct sentence; other incorrect sentences can be detected over all the text and therefore authors must review all document

Line 50: “this case is documented to show the clinical link between ruxolitinib” I consider that the word “potential” should be intrpduced between “the” and “clinical”

Line 59: ACE abbreviation was not defined in first use

5) References – mistakes can be detected in references 1, 5, 6, 7, 13, and others…, please review all references

Author Response

Response to reviewer 1

Comments and Suggestions for Authors

Q1: In this manuscript is reported a case involving the possible association between the use of Ruxolitinib and increased frequency of urinary infection caused by Klebsiella pneumoniae.

As it is known that this drug increases the susceptibility to infections and therefore the reported possible association is possible, there is no marked novelty of this work. However, I agree with the authors in that “This case is reported to add to the literature describing an increased susceptibility of patients to this often-resistant bacteria and to raise awareness about the immune modulating effects of JAK inhibitors”. Thus, I think that this document can be considered for publication.

A1: Thank you

However, several mistakes can be detected:

Q2: 1) Please correct the name of the bacterium to “Klebsiella pneumoniae” (including in italics), as mistakes can be detected over all the text.

 A2: This was done.

Q3: 2) Keywords:

-need formatting

-mistakes: “Klebesiella” and “Immune modulation”

A3: This was done.

Q4: 3) Abstract:

-lines 22/23: please correct sentence

A4: This was done.

Q5: 4-please define the abbreviation STAT

A5: This was done.

Q6: -PCV was defined twice

A6: This was done. 

Q7: 4) Main text:

Lines 42/43: please correct sentence; other incorrect sentences can be detected over all the text and therefore authors must review all document

A7: This was done, documented was professionally edited.

Q8: Line 50: “this case is documented to show the clinical link between ruxolitinib” I consider that the word “potential” should be introduced between “the” and “clinical”

A8: This was revised.

Q9: Line 59: ACE abbreviation was not defined in first use

 A9: This was defined.

Q10: 5) References – mistakes can be detected in references 1, 5, 6, 7, 13, and others…, please review all references

A10: These were revised.

Reviewer 2 Report

Dear Authors

Thank you for the time and energy for the clinical case study presentation.

I just have a few smaller items that should be corrected in the paper. Please be sure to lower case the species name. Line 80, 87,102 need correction to Klebsiella pneumoniae.

Line 87, it should be urinary instead of Urinary.

Line 87, I am confused at the use of “spp.”  You indicate “Klebsiella pneumoniae spp.” Are you indicating that there were multiple species of the same genus for each infection (mixed microbes) or a different species for each infection? The correct use may be Klebsiella sp (singular species) or spp. (plural, many different species). I would just remove spp from the sentence.

sp.= single species

spp. = multiple species

Diagram

I would suggest that you provide more information about the diagram in the text for those that do not keep track of the pathway. What do the arrows mean and briefly describe how one protein affects the other.

Thank you

Author Response

Comments and Suggestions for Authors

Dear Authors

Q1: Thank you for the time and energy for the clinical case study presentation.

A1: Thank you.

Q2: I just have a few smaller items that should be corrected in the paper. Please be sure to lower case the species name. Line 80, 87,102 need correction to Klebsiella pneumoniae.

A2: This is done.

Q3: Line 87, it should be urinary instead of Urinary.

 A3: Done.

Q4: Line 87, I am confused at the use of “spp.”  You indicate “Klebsiella pneumoniae spp.” Are you indicating that there were multiple species of the same genus for each infection (mixed microbes) or a different species for each infection? The correct use may be Klebsiella sp (singular species) or spp. (plural, many different species). I would just remove spp from the sentence.

sp.= single species

spp. = multiple species

 A4: This has been done, thank you. There was only one species and as such I got rid of spp/sp to be clear.

Diagram

Q5: I would suggest that you provide more information about the diagram in the text for those that do not keep track of the pathway. What do the arrows mean and briefly describe how one protein affects the other.

 A5: This has been done, thank you.

Thank you

Reviewer 3 Report

Ruxolitinib is a Janus 2 Kinase (JAK) inhibitor and is approved to treat polycythemia vera (PCV) and myelofibrosis. Several reports indicate that patients treated with ruxolitinib are prone to urinary tract infection (UTI), including with K. pneumoniae Hanna and colleagues in the present article reports a case of recurrent UTI with K. pneumoniae in a 70 year old patient treated with Ruxolitinib. Though this case is good for the documentation but needs additional, but I feel that authors need to add additional information, as described below, to make it appropriate for a good publication.

Major concerns:

Addition of a table summarizing all reported cases of pneumoniae mediated UTI in patient treated with ruxolitinib, will improve the value and usefulness of this manuscript. This is not evident if Hanna et al. also provided ruxolitinib to patient during the antibiotic treatment. If not, recurrent nature of infection may not be associated with the r Please describe the method used only to identify the pneumoniae. Were any other pathogenic bacteria detected?

 Minor concerns: Manuscript has numerous typos.

In the title and in the rest of the manuscript, please change “ Klebsiella Pneumonia” to “Klebsiella pneumoniae”. Please standard format of writing genera and species. At first place, write “Klebsiella pneumoniae” and at rest “ pneumoniae, in italic”. Line 46: Change “The effects of ruxolitinib has been” to “The effects of ruxolitinib have been”. Lines 53-57: Please rephrase sentences. Line 62-81: Units of all antibiotics are missing.

Author Response

Comments and Suggestions for Authors

Q1: Ruxolitinib is a Janus 2 Kinase (JAK) inhibitor and is approved to treat polycythemia vera (PCV) and myelofibrosis. Several reports indicate that patients treated with ruxolitinib are prone to urinary tract infection (UTI), including with K. pneumoniae Hanna and colleagues in the present article reports a case of recurrent UTI with K. pneumoniae in a 70 year old patient treated with Ruxolitinib. Though this case is good for the documentation but needs additional, but I feel that authors need to add additional information, as described below, to make it appropriate for a good publication.

A1: Thank you.

Q2: Major concerns:

Addition of a table summarizing all reported cases of pneumoniae mediated UTI in patient treated with ruxolitinib, will improve the value and usefulness of this manuscript.

A2: This is a great idea, it has been done see table 1.

Q3: This is not evident if Hanna et al. also provided ruxolitinib to patient during the antibiotic treatment.

A3: Yes ruxolitinib was provided for PCV along with antibiosis.

Q4: If not, recurrent nature of infection may not be associated with the drug.

A4: see above.

Q5: Please describe the method used only to identify the pneumoniae. Were any other pathogenic bacteria detected?

A5: Bacteria was identified by urinary culture, and standard staining and chemical assays, no other pathogens were detected.

 Minor concerns: Manuscript has numerous typos.

Q6: In the title and in the rest of the manuscript, please change “ Klebsiella Pneumonia” to “Klebsiella pneumoniae”.

A6: This was done.

Q7: Please standard format of writing genera and species. At first place, write “Klebsiella pneumoniae” and at rest “ pneumoniae, in italic”.

A7: This was done.

Q8: Line 46: Change “The effects of ruxolitinib has been” to “The effects of ruxolitinib have been”.

A8: This was done.

Q9: Lines 53-57: Please rephrase sentences.

A9: This was done

Q10: Line 62-81: Units of all antibiotics are missing.

A10: These were added.

Round 2

Reviewer 3 Report

Authors have justified all of my concerns, but, there are still some typos.

Klebsiella pneumoniae should be written out in full when it first appears and then abbreviated (K. pneumoniae) in the rest of the manuscript. Line 40: Please put a space in "withcellular". Table 1 & Line 90: Please change Klebsiella Pneumoniae to K. pneumoniae. Supplementary file contain draft of main manuscript. Please delete that.